# Disclosing the Interactome of Leukemogenic NUP98-HOXA9 and SET-NUP214 Fusion Proteins Using a Proteomic Approach

**DOI:** 10.3390/cells9071666

**Published:** 2020-07-10

**Authors:** Adélia Mendes, Ramona Jühlen, Sabrina Bousbata, Birthe Fahrenkrog

**Affiliations:** 1Institute of Molecular Biology and Medicine, Université Libre de Bruxelles, 6041 Charleroi, Belgium; rjuehlen@ukaachen.de (R.J.); sabrina.bousbata@ulb.ac.be (S.B.); 2Present address: Institute of Biochemistry and Molecular Cell Biology, RWTH Aachen University, 52074 Aachen, Germany

**Keywords:** SET-NUP214, NUP98-HOXA9, BioID, interactome, gene ontology, leukemia

## Abstract

The interaction of oncogenes with cellular proteins is a major determinant of cellular transformation. The NUP98-HOXA9 and SET-NUP214 chimeras result from recurrent chromosomal translocations in acute leukemia. Functionally, the two fusion proteins inhibit nuclear export and interact with epigenetic regulators. The full interactome of NUP98-HOXA9 and SET-NUP214 is currently unknown. We used proximity-dependent biotin identification (BioID) to study the landscape of the NUP98-HOXA9 and SET-NUP214 environments. Our results suggest that both fusion proteins interact with major regulators of RNA processing, with translation-associated proteins, and that both chimeras perturb the transcriptional program of the tumor suppressor p53. Other cellular processes appear to be distinctively affected by the particular fusion protein. NUP98-HOXA9 likely perturbs Wnt, MAPK, and estrogen receptor (ER) signaling pathways, as well as the cytoskeleton, the latter likely due to its interaction with the nuclear export receptor CRM1. Conversely, mitochondrial proteins and metabolic regulators are significantly overrepresented in the SET-NUP214 proximal interactome. Our study provides new clues on the mechanistic actions of nucleoporin fusion proteins and might be of particular relevance in the search for new druggable targets for the treatment of nucleoporin-related leukemia.

## 1. Introduction

The nucleoporins NUP98 and NUP214 are components of the nuclear pore complex (NPC) and belong to the group of so-called phenylalanine-glycine (FG) rich nucleoporins, which are essential for nucleocytoplasmic transport. Via their FG domains, NUP98 and NUP214 contribute to the selective and semi-permeable NPC barrier, and interact with nuclear transport receptors (NTRs) of the β-karyopherin family, thereby promoting the fast exchange of cargoes between the nucleus and the cytoplasm [1,2,3].

A number of chromosomal translocations involving the *NUP98* and *NUP214* loci are reported in acute myeloid and lymphoblastic leukemias (AML and ALL, respectively). NUP98- and NUP214-related leukemia are associated with poor overall survival [4,5,6,7], and no specific or targeted therapies are as yet available to improve prognosis. The chromosomal rearrangements of *NUP98* and *NUP214* result in their fusion with a large range of gene partners, all of which retain the FG domain of the respective nucleoporin [7,8]. The fusion of NUP98 with the homeobox protein Hox-A9 (HOXA9), NUP98-HOXA9, that results from t(7;11)(p15;p15), has been studied as the prototype for the oncogenic mechanisms governing the actions of NUP98 fusions with homeodomain (HD) proteins in AML [9]. HOXA9 is a transcription factor that regulates hematopoietic stem cell expansion and is abundantly expressed in hematopoietic precursor cells, while being progressively silenced during differentiation [10,11]. NUP214 is frequently found in conjunction with the oncogene SET, resulting from either t(9;9)(q34;q34) or an interstitial deletion at 9q34 [12,13,14]. SET-NUP214 is typically linked to ALL, and less frequently to AML [15,16]. SET is a chromatin-binding protein and an epigenetic regulator as part of the inhibitor of acetyltransferases (INHAT) complex [17,18]. Due to its role as an epigenetic modifier, SET is involved in a multitude of cellular functions, including regulation of the cell cycle, gene expression, and apoptosis [19,20,21]. NUP98-HOXA9 and SET-NUP214 share several characteristics: both form nuclear foci that accumulate endogenous proteins [22,23]; both interact with the NTR chromosome region maintenance 1 (CRM1), or exportin 1 (XPO1), and sequester cargo-loaded CRM1-nuclear export complexes to inhibit their translocation to the cytoplasm [23,24,25]. Moreover, NUP98-HOXA9 and SET-NUP214 interact with chromatin-binding proteins, such as the histone methyltransferases mixed lineage leukemia 1 (MLL1) and the disruptor of telomeric silencing 1-like (DOT1L) [26,27,28]. Association of NUP98-HOXA9 and SET-NUP214 with chromatin-bound CRM1 induces over-expression of *HOX* genes, a hallmark of unfavorable prognosis in leukemia [10,29,30]. The full landscape of NUP98-HOXA9 and SET-NUP214 interactors, however, has not yet been determined.

In recent years, advances in enzyme-mediated protein labelling became a powerful approach to study specific protein–protein interactions (PPIs) [31,32,33]. Proximity-dependent protein biotinylation (BioID) is an enzyme-mediated protein labelling approach that uses a modified version of the *Escherichia coli* (*E. coli*) biotin ligase BirA (BirA^R118G^), which has a lower affinity than wild type BirA for biotinoyl-adenylate (bio-AMP), the active form of biotin that can bind lysine residues [31,34]. Thus, when expressed in-frame with a protein of interest, BirA^R118G^ biotinylates proximal proteins, which can then be purified by streptavidin pulldown and further identified by mass spectrometry [32]. In contrast to other interaction assays, which may be regarded as a snapshot of PPIs at the moment of cell lysis, BioID interrogates both stable and transient PPIs in living cells, thus providing a broader picture of protein interactors. Here, we used a modified BioID approach to study the proximal interactome of NUP98-HOXA9 and SET-NUP214. We identified further common associated partner as well as discrete fusion protein-specific interactors in the environment of NUP98-HOXA9 and SET-NUP214.

## 2. Materials and Methods

All experiments were carried out at room temperature (RT) unless otherwise specified.

### 2.1. Plasmids

*pcDNA3.1 MCS-BirA(R118G)-HA* was a gift from Dr. Kyle Roux (Addgene plasmid # 36047; [31]) and *BirA(R118G)-HA* destination vector from Dr. Karl Kramer (Addgene plasmid # 53581). For the cloning of *SET-NUP214*, total RNA was extracted from LOUCY cells, which carry del(9)(q34.11q34.13) resulting in the *SET-NUP214* fusion transcript [35]. The coding sequence of *SET-NUP214* was cloned into the *pcDNA3.1 MCS-BirA(R118G)-HA* vector, as described in Appendix B. The *NUP98-HOXA9-BirA^R118G^* construct was generated by Gateway^®^ cloning. The coding sequence of *NUP98-HOXA9* was first subcloned from *pEGFP-NUP98-HOXA9* [22] into the *pENTR/TOPO* vector using the TOPO^®^ TA Cloning Kit (Invitrogen, Merelbeke, Belgium) to generate the *pENTR/NUP98-HOXA9* Gateway^®^ entry vector. The *NUP98-HOXA9* sequence was then subcloned into the *BirA(R118G)-HA* destination vector using the Gateway ™ LR Clonase™ enzyme mix (Invitrogen).

### 2.2. Cell Lines and Transfections

HCT-116 cells were a gift from Dr. Denis Lafontaine (Institute of Molecular Biology and Medicine, Université Libre de Bruxelles, Charleroi, Belgium). HCT-116 cells were cultured in McCoy’s 5A medium (LONZA^TM^ BioWhittaker^TM^, Verviers, Belgium), supplemented with 10% FBS and 1% penicillin/streptomycin (P/S, GIBCO, Invitrogen), and cultivated in a humidified incubator at 37 °C with 5% CO_2_ atmosphere. HCT-116 cells were transfected using the jetPRIME^®^ transfection reagent (Polyplus transfection^®^, Illkirch, France). Briefly, plasmids were mixed with transfection reagent at a 1:2 (*w*/*v*) ratio and incubated for 40 min. Transfection mixes were added to the cell culture and incubated for 24 h. Next, the culture medium was replaced by fresh medium containing 50 µM biotin (Sigma–Aldrich, Overijse, Belgium) and biotinylation was induced for an additional 24 h. Cells were tested for mycoplasma contamination on a regular basis.

### 2.3. Immunofluorescence

Cells were grown on polylysine-coated glass coverslips and fixed in 2% formaldehyde for 15 min, washed three times for 10 min with PBS, and permeabilized with PBS/2% BSA/0.1% Triton X-100 for 10 min. Cells were washed twice for 10 min in PBS/2% BSA, incubated with Streptavidin-Alexa Fluor ™ 488 conjugate (dilution 1:1000; Invitrogen) or anti-HA antibody (clone 12CA5, dilution 1/50, supernatant of a mouse hybridoma cell line) for 1 h and washed twice in PBS/2% BSA/0.1% Triton X-100. Cells incubated with anti-HA were then incubated with goat anti-mouse IgG Alexa Fluor™ 488 (dilution 1:1000; Invitrogen) for 1 h and washed twice for 10 min with PBS. All cells were mounted with Mowiol-4088 containing DAPI (1 μg/ml) and stored at 4 °C until viewed. Cells were imaged using a Zeiss LSM-710 confocal laser-scanning microscope (Zeiss, Oberkochen, Germany). Images were recorded using the microscope system software and processed using ImageJ v.1.52t (http://imagej.nih.gov) and Inkscape 0.92 Software (http://www.inkscape.org).

### 2.4. Pulldown of Biotinylated Proteins

2 × 10^6^ cells were plated in a 10 cm^2^ cell-culture dish, grown for 24 h and subsequently processed for transfection and biotinylation induction as described above. Cells were lysed in lysis buffer (50 mM Tris-HCl, pH 7.8, 150 mM NaCl, 1mM EGTA, 1.5 mM MgCl_2_, 0.4% sodium dodecyl sulfate (SDS), 1 µl/ml benzonaze [25 U/ml], 1% Nonidet-P40, and protease inhibitor cocktail tablets (Roche, Basel, Switzerland). Bradford assay was used to determine protein concentration and 500 μg of protein were incubated with 50 µl of SeraMag™ magnetic Streptavidin-coated beads ([10 mg/ml]; GE Healthcare, Chicago, Illinois, USA). Protein-beads incubation and recovery of biotinylated proteins were carried out as detailed in Appendix B. The entire eluate containing biotinylated proteins was subjected to sodium dodecyl-sulfate polyacrylamide gel electrophoresis (SDS-PAGE) and Western blotting. Proteins were detected using horseradish-conjugated streptavidin (HRP-Strep; Thermo Fischer Scientific, Merelbeke, Belgium). For protein identification by mass spectrometry, the same amount of protein was used. After incubation of whole protein extract with streptavidin-coated magnetic beads, samples were resuspended in on-bead tryptic digestion buffer (20 mM Tris-HCl, pH 8.0, 2 mM CaCl_2_) and processed for large-scale analysis by tandem mass spectrometry (LC-MS/MS).

### 2.5. Mass Spectrometry

Proteins were digested on the beads with 1 µg of trypsin (Promega, Leiden, Netherlands) at 37 °C for 4 h while spinning at 161× *g*. Beads were removed, an additional 1 µg of trypsin was added, and proteins were further digested at 37 °C, overnight. The resulting peptide mixture was purified using OMIX C18 pipette tips (Agilent, Santa Clara, California, USA). The purified peptides were dried completely and re-suspended in 20 µl loading solvent (0.1% TFA in water/ acetonitrile, 2/98 (v/v)) of which 5 µl were injected for LC-MS/MS analysis on an Ultimate 3000 RSLCnano ProFLow system connected online to a Q Exactive HF mass spectrometer (Thermo, Waltham, MA, USA). Peptide trapping and elution and mass spectrometer operation details are described in Appendix B (Sample injection and mass spectrometer operation).

### 2.6. Data Analysis

Data analysis was performed with MaxQuant (version 1.6.3.4; Max Planck Institute of Biochemistry, Germany [36]) using the Andromeda search engine with default search settings including a false discovery rate (FDR) set at 1% on both the peptide and protein level. Spectra were searched against the human UniProt Tax ID: 9606 proteins in the UniProt/Swiss-Prot reference database (database release version of January 2019, www.uniprot.org) supplemented with the BirA fusion proteins, as detailed in Appendix B (*Data Analysis*). Further data analysis was performed using the Perseus software (version 1.6.2.1, Max Plank Institute of Biochemistry, Germany) after loading the protein groups file from MaxQuant [37]. First, proteins identified by site and reverse database hits and potential contaminants were removed. Label-free quantification (LFQ) values were then used to normalize protein abundance among NUP98-HOXA9-BirA^R118G^ (NHA9-BioID) and SET-NUP214-BirA^R118G^ (SN214-BioID) proximal interactors, relative to BirA^R118G^ alone (control). Given the potential differences in the expression of the BioID proteins (due to transient transfection), we performed an internal normalization to calculate the LFQ value of each individual protein that results from differences in NHA9-BioID and SN214-BioID protein expression. Subsequently, an external normalization to calculate the LFQ value of each individual protein relative to the control BirA^R118G^ (NHA9-BioID/BirA^R118G^ and SN214-BioID/BirA^R118G^) was carried out. The results were then converted to log2 and expressed as fold change (F.C.) relative to the control (Appendix A).

### 2.7. Gene Ontology and Pathway Analysis

Gene ontology (GO) of NHA9-BioID and SN214-BioID proximal interactors was performed by the GO consortium-associated Protein Analysis Through Evolutionary Relationships (PANTHER) Classification System (version 14.1.), available online (www.pantherdb.org) using the default parameters: Fisher’s exact test, with the Benjamini–Hochberg FDR correction for multiple testing and the background reference list: *Homo sapiens* whole genome [38,39,40]. Clustered enrichment analysis was performed with the Cytoscape software (v3.7.1) plugin ClueGO (v2.5.5), to calculate enrichment of terms as right-sided tests based on the hypergeometric distribution [41]. ClueGO uses Cohen’s kappa statistics to link the terms in the network and to determine the association strength between the terms, which is an indication of term overlapping to define functional clusters [41,42]. For the analysis of proximal interactors of NUP98-HOXA9 and SET-NUP214, the Kyoto Encyclopedia of Genes and Genomes (KEGG) and REACTOME Pathways together with GO vocabularies (GO Biological Processes (GOBP), GO Molecular Function (GOMF), and GO Cellular Compartments (GOCC)) were used, and functional representations of non-redundant and over-represented terms within the input protein sets were generated. Human Ensemble Gene ID identifiers (ENSG IDs) were mapped to the selected annotations (GO, updated on 27th February, 2019, KEGG, updated on 19th December, 2019, and REACTOME pathways, updated on 19th December 2019). The following settings were used during enrichment analysis: *p*-value cut off 0.01, and Bonferroni step-down correction for multiple comparisons.

### 2.8. Screening for Nuclear Export Signals

The presence of classical NESs was assessed using the online available software NES finder 0.2 (http://research.nki.nl/fornerodlab/NES-Finder.htm) and LocNES (http://prodata.swmed.edu/LocNES/LocNES.php), with default software parameters [43,44].

## 3. Results

### 3.1. Subcellular Distribution of BioID Fusion Proteins and Biotinylation Induction

To validate the subcellular localization of the BioID fusion proteins, we first examined the localization of NHA9-BioID, SN214-BioID, and control BirA^R118G^, after biotinylation was induced. As shown in Figure 1A, NHA9-BioID localized to the nucleus in a distinctive punctate pattern, whereas SN214-BioID formed intranuclear foci and localized to the nuclear rim. The localization of the two BioID fusion proteins is similar to the respective GFP-fusion proteins of NHA9 and SN214 (Figure 1B; [22,45]). BirA^R118G^ was found throughout the entire cell (Figure 1A). Streptavidin labeling revealed the same distribution pattern as HA- and GFP-NHA9 and -SN214, indicating the presence of biotin (Figure 1C). Western blot analysis furthermore showed that endogenous proteins were biotinylated in whole cellular extracts of NHA9-BioID and SN214-BioID transfected cells (Figure 1D, bound fraction), in contrast to BirA^R118G^ transfected cells. Of note, the NHA9-BioID fraction showed a strong enrichment of a specific band above 100 kDa, likely corresponding to the fusion protein itself. Due to the lower intensity of the SN214-BioID fraction, biotinylated SET-NUP214 cannot reliably be allocated, but might correspond to the band appearing below the 250-kDa band. For BirA^R118G^ no specific enrichment was observed, given the unspecific nature of biotinylation mediated by the biotin ligase alone.

### 3.2. Identification of Known Proximal Interactors of NUP98-HOXA9 and SET-NUP214

Having validated the correct subcellular localization and the reliability of NHA9-BioID and SN214-BioID, we then carried out a mass-spectrometric analysis of the biotinylated proteins co-purifying with the respective BioID fusion protein. We first screened for candidate proximal interactors and identified several known binding partners of NUP98-HOXA9 and SET-NUP214. These were the protein nuclear export factor CRM1 [24,29], the mRNA export factor RAE1 [46], the transcription factor NFAT5 [24], the histone methyltransferase MLL1 [28], and the histone deacetylase HDAC1 [47] for NUP98-HOXA9 (Table 1). We did not find the acetyltransferase CREB binding protein/p300 (CBP/p300; [48]) nor WDR5, a component of the non-lethal specific (NSL)-histone modifying complex [27], co-purifying with NHA9-BioID, which might be due to the fact that both genes are mutated in HCT-116 cells [49,50]. Among known SET-NUP214 interactors (Table 2), we identified the nuclear RNA export factor 1 (NXF1/TAP; [25]), CRM1 [23,25], and nucleoporin NUP62 [23,45].

A major limitation of our BioID approach is the use of transient transfection instead of stable, inducible expression of the BioID fusion proteins. Due to its smaller size, the transfection efficiency of BirA^R118G^ is much higher as for the larger NHA9-BioID and SN214-BioID constructs. We therefore considered the rate of biotinylation by BirA^R118G^ higher than by the fusion proteins. Moreover, BirA^R118G^ is a promiscuous biotin ligase, which also likely results in higher biotinylation levels in the control. To control for these differences in biotinylation, we employed a two-step normalization strategy that accounted for differences between BirA^R118G^, NHA9-BioID, and SN214-BioID protein expression (Appendix A). To further validate our BioID approach, next we performed Gene ontology (GO) analysis of proteins exclusively co-purifying with the BirA^R118G^ control to determine unspecific interactors of BirA^R118G^-mediated biotinylation [29], as described in the Methods section. Appendix A summarizes the list of proteins that were exclusively biotinylated by BirA^R118G^ and their respective subcellular localization. No significant enrichment in any of the GO categories (i) biological processes (GOBP), (ii) molecular function (GOMF), and (iii) cellular components (GOCC) was found, suggesting that biotinylation of these proteins is of an unspecific nature, as previously reported [29].

Moreover, given the differences in transfection efficiency between the BirA^R118G^, NHA9-BioID, and SN214-BioID constructs, some potential candidates might have disappeared, because known interactors, such as CRM1 and others, were more strongly biotinylated by BirA^R118G^ as compared to NHA9-BioID and SN214-BioID. This in consequence explains the negative fold change (F.C.) values (Table 1). High normalized label-free quantification values (LFQ_NORMALIZED_; see *Data Analysis*, Section 2.6) indicated that these proteins, after pull-down, are more enriched with NHA9-BioID or SN214-BioID, albeit being biotinylated by BirA^R118G^, meaning that their interaction with the fusion proteins is of a rather specific nature.

### 3.3. Identification of Novel Proximal Interactors of NUP98-HOXA9

Next, we employed a two-step normalization strategy to evaluate NHA9-BioID and SN214-BioID proximal interactors relative to BirA^R118G^ to account for differences in protein expression levels due to the transient expression of the BioID fusion proteins (Appendix A). In doing so, we obtained 131 proteins that were at least 50% more abundant for NHA9-BioID when compared to BirA^R118G^ (Appendix A, fold change ≥0.6). GOCC analysis revealed that these NHA9-BioID proximal interactors located to both cytoplasmic and nuclear compartments with a general enrichment of chromatin-associated proteins and particularly the transcription factor AP-1 complex (Figure 2A). Cytoskeleton-related proteins were also significantly overrepresented in the pool of NHA9-BioID proximal interactors, namely proteins from the spindle pole, actin stress fibers, and centrosome (Figure 2A,B). GOMF analysis showed that NUP98-HOXA9 proximal interactors are significantly enriched in DNA binding proteins that target the promoters of genes transcribed by RNA polymerase II (RNAPII; Figure 2C).

For a systematic analysis of the landscape of the NUP98-HOXA9 proximal interactome, next we performed clustered pathway analysis (Figure 3). Here, the most significantly enriched functional clusters included proteins involved in RNA processing and the RNAPII transcriptional machinery (Figure 3, Appendix A). Moreover, proximal NHA9-BioID interactors clustered in functional groups associated with the expression of genes from key signaling pathways frequently dysregulated in cancer. These included estrogen receptor (ER) and MAPK signaling pathways, and p53-mediated transcription of DNA damage repair genes (Figure 3A, Appendix A; [51,52,53]). Further, β-catenin-mediated transcription may be negatively regulated by NUP98-HOXA9, suggesting a dysregulation of the Wnt signaling pathway. Interestingly, some members of the Wnt signaling pathway, i.e., the segment polarity protein disheveled homolog DVL-1, the trinucleotide repeat-containing gene 6A protein (TNRC6A), and the transducing-like enhancer protein, isoforms 1-4 (TLE 1-4), were exclusively biotinylated by NHA9-BioID, further supporting the hypothesis that NUP98-HOXA9 affects this signaling pathway. Altogether, GO and clustered pathway analyses of NUP98-HOXA9 suggest that transcription dysregulation is a major defect in NUP98-HOXA9 driven leukemia.

### 3.4. NUP98-HOXA9 Proximal Interactors are Enriched in Nuclear Export Signal (NES) Containing Proteins

NUP98 is essential for CRM1-mediated nuclear export and NUP98 chimeras have been shown to affect CRM1-mediated nuclear export, to an extent, however, that is not fully understood [7,54]. To obtain a deeper insight into the impact of NUP98-HOXA9 on CRM1-mediated nuclear export, we screened the list of NUP98-HOXA9 proximal interactors for the presence of classical NESs using two algorithms: NES finder 0.2 and LocNES [43,44]. Both algorithms revealed that the majority of NHA9-BioID proximal interactors exhibit at least one classical NES motif (NES+ proteins, Figure 4A, Appendix A). Given that the LocNES software has previously been shown to identify a higher rate of false positive NESs, we applied GO analysis for predicted NES+ interactors obtained by the NES Finder 0.2 software [55]. As shown in Figure 4B, this revealed a significant enrichment in proteins from cytoplasmic ribonuclear granules and in proteins from the microtubule organizing center. Conversely, NES- proximal interactors of NUP98-HOXA9 were significantly enriched in proteins implicated in transcription regulation, namely the ß-catenin/TCF complex and the transcription factor AP-1 complex (Figure 4B).

### 3.5. Identification of Novel Proximal Interactors of SET-NUP214

Our LFQ normalization strategy produced a list of 1125 proteins enriched in SN214-BioID relative to the BirA^R118G^ control (Appendix A). To disclose the SET-NUP214 proximal interactome, we performed GO enrichment analysis of the SN214-BioID fraction using the same fold-change threshold of ≥0.6 and statistical parameters as for NHA9-BioID (Figure 5, Appendix A). For clustered pathway analysis, we applied a conservative approach to reduce redundancy of the functional network given the elevated number of potential SN214-BioID interactors. Figure 6 shows the functional groups that were significantly represented, obtained with the following statistical parameters: *p* <0.01, GO levels: 13-15 (ClueGO default: 3-8), threshold kappa score of 0.5 (ClueGO default 0.4), thus increasing the threshold for group overlapping, and the minimum gene number corresponding to percentage of genes of 40/4% (ClueGO default 3/4%).

GOCC (Figure 5A and Appendix A) and GOBP (Figure 5B and Appendix A) analysis produced a list of cytoplasmic and nuclear structures and processes likely to be affected by SET-NUP214, such as mRNA processing, intracellular transport, viral transport, and transcription (Figure 5B, Appendix A) [48,50,51,52,53]. Consistently GOMF analysis (Figure 5C) unveiled an enrichment in proteins with GTP- and GDP-binding activity, as well as mRNA and DNA binding proteins. Moreover, proteins involved in neutrophil degranulation were enriched with SET-NUP214 (Figure 5B), establishing a direct link between SET-NUP214 and immune regulation. GO also showed an enrichment in nucleolar and ribosomal proteins (Figure 5A and Appendix A), suggesting an effect of the fusion protein on translation. It further revealed that mitochondrial proteins were enriched with SN214-BioID, especially proteins from the respiratory chain complexes and proteins involved in mitochondrial translation (Figure 5A, Appendix A), suggesting involvement of the fusion protein in cell metabolism through an effect on mitochondria. In line with the GO results, clustered pathway analysis of proximal SET-NUP214 interactors revealed an enrichment in proteins involved in amino acid metabolism, translation and infection, and proteins involved in virus biology, such as transcription, transport, and interaction with host cells (Figure 6, Appendix A). Clustered pathway analysis further supports GO findings of an interplay between SET-NUP214 and mitochondrial proteins (Figure 6B, Appendix A). Moreover, the results suggest an association of SET-NUP214 with several transcription factors and point to a possible effect on *TP53*-mediated transcription (Figure 6A,B, Appendix A). Finally, our clustered pathway analysis reinforces the link between SET-NUP214 and immunity, with two distinct, yet overlapping, immunity-related clusters that link the fusion protein to immune system regulation, and more specifically to neutrophil activation (Figure 6A, Appendix A).

We identified a total of 47 proteins that were enriched with both NHA9-BioID and SN214-BioID, of which a vast majority presented at least one putative classic NES (Appendix A). Furthermore, GO analysis showed that the pool of shared proximal interactors is enriched in proteins associated with microtubule cytoskeleton, suggesting that SET-NUP214, like NUP98-HOXA9, might be involved in microtubule organization (Appendix A).

## 4. Discussion

We used a modified BioID approach to study the proximal interactome of NUP98-HOXA9 and SET-NUP214. We compared the pool of proteins biotinylated by NHA9-BioID and SN214-BioID, respectively, with the pool of proteins biotinylated by BirA^R118G^, using a normalization strategy that accounts for differences in NHA9-BioID, SN214-BioID and BirA^R118G^ expression due to their transient transfection. After validating the expression, cellular distribution, and the capacity of protein biotinylation of all three BioID proteins, we performed gene ontology (GO) and clustered pathway enrichment analysis of the proximal interactors of NHA9-BioID and SN214-BioID. As expected, we observed that BirA^R118G^ alone biotinylates endogenous proteins in an unspecific manner [31]. The respective proximal interactomes of NHA9-BioID and SN214-BioID were, somewhat surprisingly, enriched in cytoplasmic proteins, which appears inconsistent with the nuclear localization of the fusion proteins. This might be explained by the fact that mitosis in vertebrates involves nuclear envelope breakdown and mixing of the cytoplasmic and nuclear protein content [56]. At the end of mitosis, proteins must be appropriately segregated between the nucleus and the cytosol. This process depends on transport factors, including CRM1, which alone is predicted to transport one fourth of the entire proteome [57]. As shown by us and others, NUP98-HOXA9 and SET-NUP214 sequester CRM1 nuclear export complexes, resulting in the nuclear accumulation of proteins and RNPs [23,24,25]. Thus, it is possible that in cells expressing NUP98-HOXA9 and SET-NUP214 fusion proteins protein segregation after mitosis is affected, and that otherwise exclusively cytoplasmic proteins are retained in the nucleus.

The results from GO and pathway enrichment analysis suggest that both NUP98-HOXA9 and SET-NUP214 interact with major regulatory proteins, such as members of the RNAPII complex, ribosomal proteins, and transcription factor complexes, suggesting a widespread effect on gene expression (Figure 2, Figure 3, Figure 4, Figure 5 and Figure 6). NUP98-HOXA9 and SET-NUP214 form dynamic nuclear foci that accumulate endogenous proteins [23,46]. These structures may represent factories that bring transcriptional and epigenetic regulators into close proximity to promote gene expression changes [29,30,58]. Interestingly, our analyses suggest that p53-mediated transcription is a common target of NUP98-HOXA9 and SET-NUP214. The association of the fusion proteins with p53 might occur via their NUP98 and SET portions, respectively. Both proteins regulate *TP53*-mediated transcription, namely of the cyclin-dependent kinase inhibitor p21 (p21^CDKN1A^), a DNA damage response and cell cycle regulator [59,60,61]. p53 dysregulation is frequent in both ALL and AML, and loss of p53 function was previously reported to promote AML progression in a NUP98-HOXD13 mouse model, suggesting that it might contribute to NUP98-related leukemia [62,63,64]. Nevertheless, despite the direct binding between SET and p53, the status of p53 signaling in SET-NUP214 leukemia has not been studied.

Changes in gene expression by NUP98-HOXA9 might result from its interaction with major transcription factors, such as TFAP2A (Appendix A) and the AP-1 transcription factor complex (Figure 2 and Figure 3), and proteins from cytoplasmic RNP granules (Figure 2 and Figure 3), where mRNA is processed for translation or targeted for degradation [47,65,66,67,68]. Chromatin immunoprecipitation (ChIP) experiments showed that NUP98-HOXA9 (but not NUP98 nor HOXA9) binds at genomic regions that are adjacent to the TFAP2A sequence recognition motif, further supporting a potential interaction between the fusion protein and this transcription factor [47]. TFAP2A regulates the transcription of several developmental genes and has been reported to promote *HOX* gene upregulation of clustered *HOX* genes in AML [69]. Moreover, members of Wnt, MAPK, and ER signaling pathways are enriched with NHA9-BioID (Figure 3). Among them, the DVL-1, TLE, isoforms 1–4, and TNRC6A proteins, which are common to Wnt and ER signaling, were exclusively found in the NHA9-BioID pool of biotinylated proteins, reinforcing the idea of a specific effect of NUP98-HOXA9 in these two signaling pathways. In line with our findings, previous work showed that in NUP98-HOXA-transduced hematopoietic stem cells, Wnt and ER signaling pathways were dysregulated, which correlated with cellular transformation [67].

Most of the NHA9-BioID proximal interactors are predicted to have at least one NES, supporting the idea that the biological consequences of CRM1 inhibition by the fusion protein might depend on the type of cargoes that become trapped in the nucleus. NES+ NHA9-BioID proximal interactors were enriched in cytoplasmic RNP granules and microtubule organizing center (MTOC)-associated proteins (Figure 4B). As a MTOC docking protein, CRM1 facilitates microtubule nucleation at the NPC periphery [70]. Several cell-death-related processes, such as autophagy and apoptosis, which are activated under stress, rely on the microtubule system [71,72]. Still, the biological implications of the nuclear retention of MTOC-related proteins, which has been reported in cancer cells, have remained unclear [73,74].

Among SET-NUP214 proximal interactors, our results revealed an unexpected enrichment of mitochondrial proteins. This enrichment might be explained by changes in the communication between the mitochondria and the nucleus (anterograde/retrograde transport) that may be imposed by SET-NUP214 at the NPC. The molecular determinants of anterograde and retrograde transport are still unclear, but current evidence shows that some mitochondrial proteins also have functions in the nucleus, and that nuclear translocation of mitochondrial proteins is an emerging mitochondrial signaling pathway [75,76]. This system relies on the microtubule network that proteins and even entire organelles use to move from and towards the vicinity of the NE, respectively [77]. Given the recently reported MTOC function of CRM1, it is tempting to hypothesize that SET-NUP214 might dock CRM1 at the cytoplasmic side of the NPC and support its MTOC function, thereby promoting the accumulation of mitochondrial proteins (or even entire mitochondria) in the periphery of the NPC. The possibility that SET-NUP214 is related to mitochondria dysregulation is supported by the observation that patients with SET-NUP214 T-related ALL are resistant to glucocorticoid (GC) therapy, which induces a metabolic shift from glycolysis to oxidative phosphorylation (OXPHOS; [20,78]). The AML-associated DEK-NUP214 fusion protein, which has the same NUP214 portion as SET-NUP214, also promotes a shift from glycolysis to OXPHOS [79]. Yet, in a recent proteomics report, no mitochondrial proteins were reported in the DEK-NUP214 interactome and dysregulation of the mammalian target of rapamycin (mTOR) was assumed as the main cause of metabolic shift [80].

Among SN214-BioID interactors, eight proteins (i.e., RAB6A, PCBD1, RPTOR, RIN1, NRAS, GCC2, LYN, and CRKL) were common to the DEK-NUP214 interactome (Appendix A; [80]). The association of these proteins with DEK-NUP214 was proposed to promote activation of several cancer associated pathways, such as AKT/mTOR, Src family kinase (SFK), ABL1, and c-MYC pathways [81,82,83,84]. Further studies will be necessary to confirm the same pathway profile activation by SET-NUP214. Nevertheless, given the structural similarities of the two chimeras, it is reasonable to expect a significant overlap in the biological processes that are affected by both fusion proteins [12,14,85].

Despite the identification of the above-mentioned numerous NUP98-HOXA9 SN214-BioID interactors, our approach possibly falls short of others due to the promiscuous biotinylation activity of BirA^R118G^. Although our normalization strategy accounts for differences in the expression of the BioID fusion proteins, we observed that some known NUP98-HOXA9 and SET-NUP214 binding partners, such as CRM1 (for both fusion proteins) and MLL1 (for NUP98-HOXA9), were not strongly enriched in the NHA9-BioID and SN214-BioID fractions relative to BirA^R118G^ alone. We hypothesize that the same may occur with other NUP98-HOXA9 and SET-NUP214 interactors and for that reason we recognize that our BioID approach might miss relevant candidate binding partners. Nevertheless, the unspecific protein biotinylation by BirA^R118G^ alone, and the identification of several known binding partners enriched with NHA9-BioID and SN214-BioID, reinforce our approach in the study of NUP98-HOXA9 and SET-NUP214 proximal interactome.

## 5. Conclusions

Overall, our report provides new data on the landscape of potential binding partners of nucleoporin fusion proteins, and suggests novel cellular processes and signaling pathways that may be affected by NUP98-HOXA9 and SET-NUP214. Although we identified several previously validated binding partners of both fusion proteins, experimental validation by gold-standard experimental methods, such as protein immunoprecipitation, is necessary to confirm the predicted interaction candidates. Our work unveils, for the first time, putative new players in nucleoporin-related leukemia, and provides the basis for a new understanding of the biological actions of nucleoporin fusion proteins. Our findings may be of particular relevance in the search for new druggable targets, such as the MAPK and ER pathways, that might be explored in the development of specific therapies for NUP98 and NUP214 leukemia.

## Figures and Tables

**Figure 1 cells-09-01666-f001:**
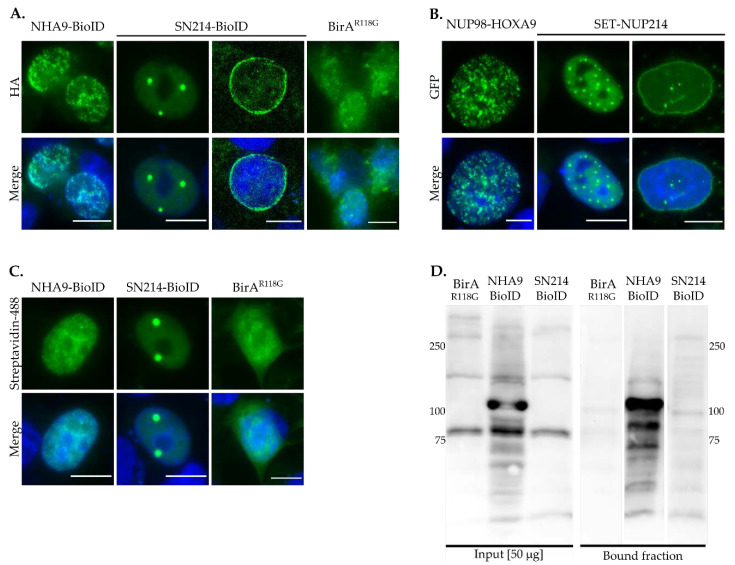
Proximity-dependent biotin identification (BioID) fusion protein expression and biotinylation of endogenous proteins. (**A**) Localization of NHA9-BioID, SN214-BioID, and BirA^R118G^ was evaluated by immunostaining with anti-HA antibody. (**B**) Localization of GFP-tagged NUP98-HOXA9 and SET-NUP214 was evaluated by green fluorescent protein (GFP) fluorescence. NHA9-BioID and SN214-BioID exhibit the same distribution pattern in nuclear foci and at the nuclear envelope as their GFP-tagged counterparts. (**C**) Detection of protein biotinylation by Streptavidin-488. Cells were transfected with NHA9-BioID, SN214-BioID, and BirA^R118G^ and probed with Streptavidin-Alexa Fluor™ 488 conjugate. Shown are representative confocal images. DNA was visualized with DAPI (blue). Scale bars, 10 µm. (**D**) For detection of protein biotinylation by NHA9-BioID, SN214-BioID, and BirA^R118G^, corresponding cell lysates were enriched on Streptavidin-coated magnetic beads and whole protein lysates and the bound fractions were analyzed by immunoblotting. Note, virtually no specific bands were detected in the bound fraction of BirA^R118G^, in contrast to the bound fractions of NHA9-BioID and SN214-BioID, which exhibited patterns of differentially biotinylated proteins.

**Figure 2 cells-09-01666-f002:**
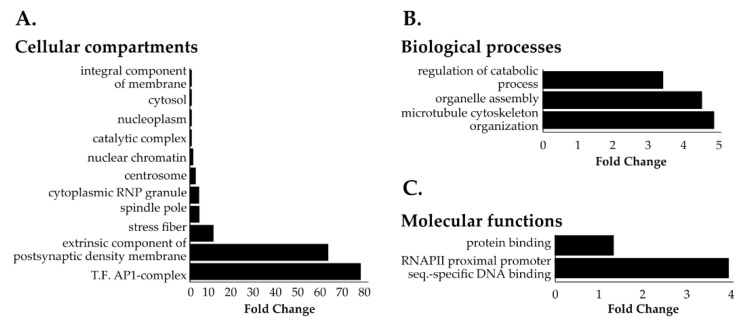
Gene ontology (GO) of NHA9-BioID proximal interactors. Most represented (**A**) cellular compartments (GOCC), (**B**) biological processes (GOBP), and (**C**) molecular functions (GOMF) among NHA9-BioID proximal interactors. Statistical analysis of the overrepresented proteins in the NHA9-BioID fraction (total 131 proteins) was performed with the PANTHER classification online software (v14.1) using Fisher’s exact test. Results are displayed for FDR *p* < 0.05.

**Figure 3 cells-09-01666-f003:**
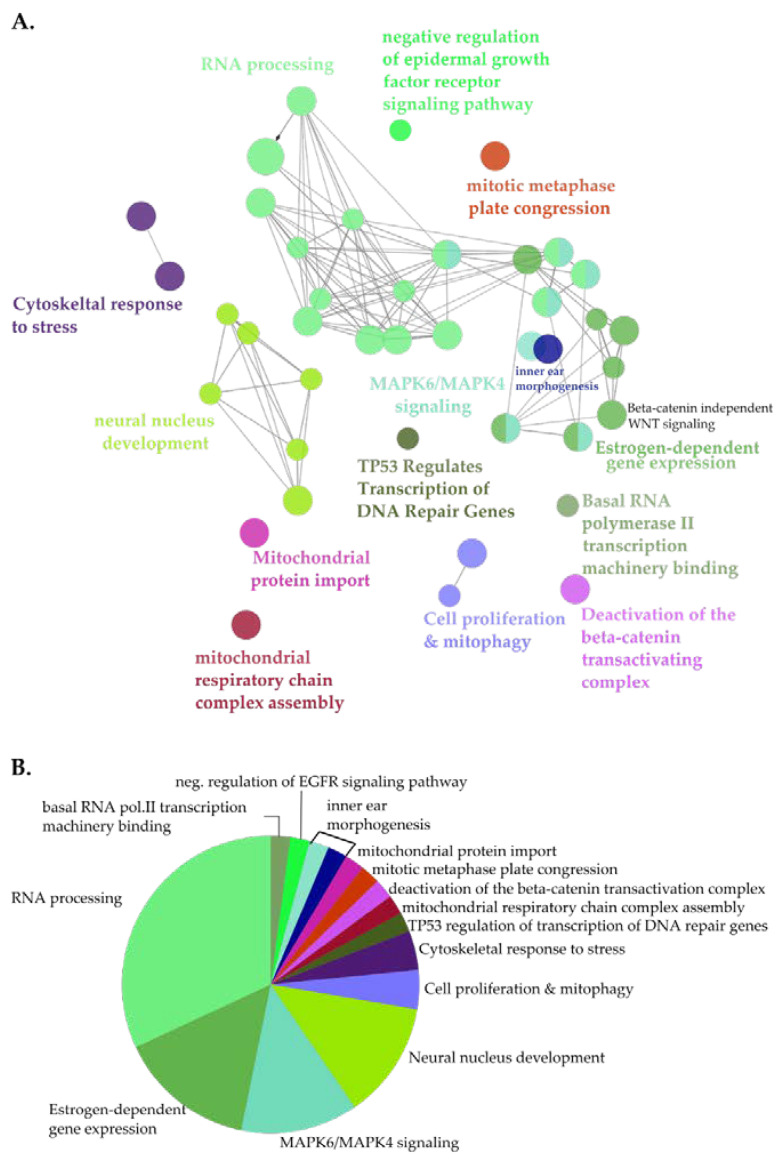
Clustered pathway analysis of NHA9-BioID proximal interactors. (**A**) Functionally grouped network of NHA9-BioID proximal interactors. Nodes correspond to functional clusters, resulting from term grouping based on their overlapping level. (**B**) Overview pie-chart showing functional groups. Statistical analysis was performed using the Cytoscape plugin ClueGo (v.2.5.5) using the hypergeometric test and the following parameters: *p* < 0.01, kappa-score (k) = 0.4; (min/%) genes = 3/4%, GO tree levels: 3-11. Ontology databases: GOBP, GOCC, and GOMF, Reactome Pathways and KEGG.

**Figure 4 cells-09-01666-f004:**
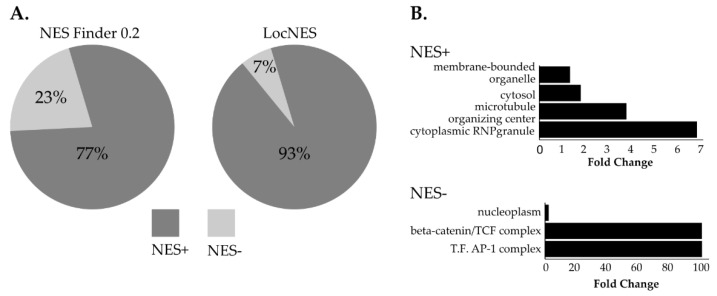
Screening of nuclear export signals (NESs) in NHA9-BioID proximal interactors. The amino acid sequences of the 131 NHA9-BioID proximal interactors were analyzed by two different algorithms (NES Finder 0.2 and LocNES) to evaluate the presence of classical NES. (**A**) The results from both algorithms show that the majority of NHA9-BioID proximal interactors have at least one putative classical NES, with a higher percentage of NES+ proteins determined by the LocNES algorithm. (**B**) GO analysis of NES+ and NES- proteins identified by NES Finder 0.2 shows an overrepresentation of proteins from cytoplasmic RNP granules and the microtubule organizing center. NES- proteins are mostly nucleoplasmic and are associated with the ß-catenin and the AP-1 transcription factor complexes. Statistical analysis was performed with the PANTHER classification online software (v14.1), using Fisher’s exact test. Results are displayed for FDR *p* < 0.05.

**Figure 5 cells-09-01666-f005:**
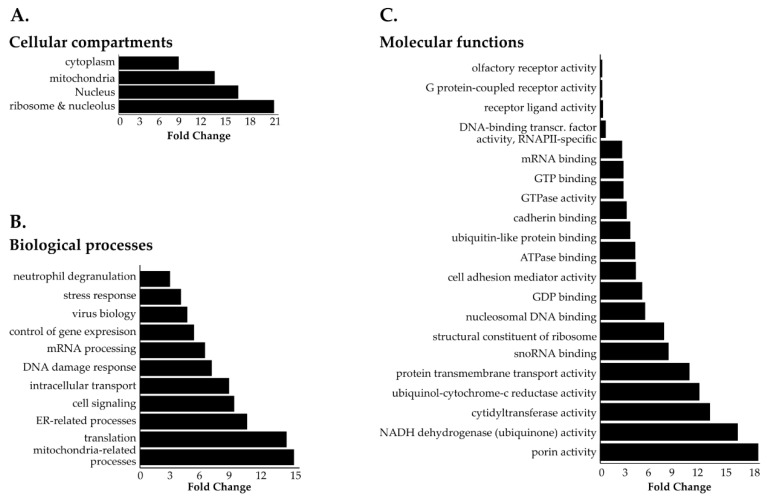
Gene ontology (GO) of SN214-BioID proximal interactors. Most represented (**A**) cellular compartments (GOCC), (**B**) biological processes (GOBP), and (**C**) molecular functions (GOMF) among SN214-BioID proximal interactors. Statistical analysis of the overrepresented proteins in the SN214-BioID fraction (total 1125 proteins) was performed with the PANTHER classification online software (v14.1) using Fisher’s exact test. Results are displayed for FDR *p* < 0.05. Summarized graphic representation of the significantly enriched GO terms among SN214-BioID proximal interactors. The detailed results of GOBP and GOCC analysis can be consulted in Appendix A.

**Figure 6 cells-09-01666-f006:**
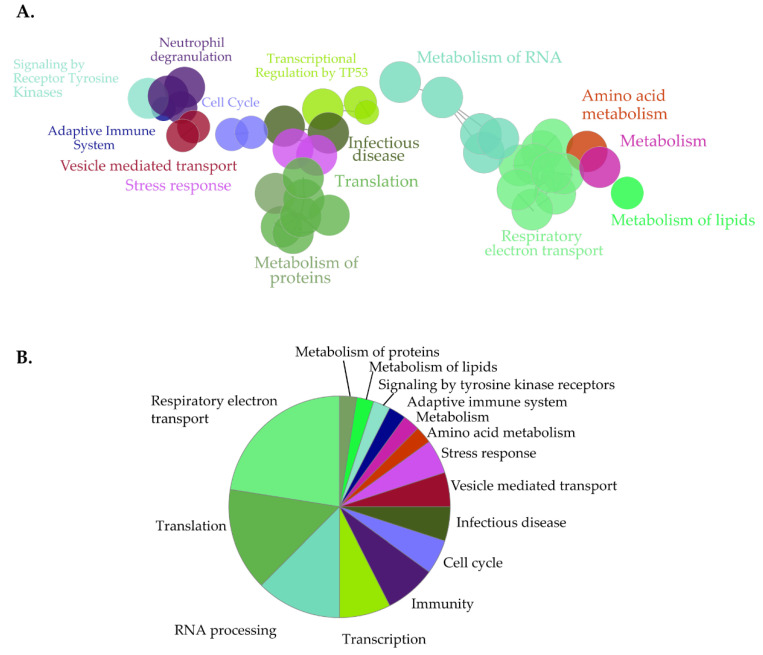
Clustered pathway analysis of SN214-BioID proximal interactors. (**A**) Functionally grouped network of SN214-BioID proximal interactors. Nodes correspond to functional clusters resulting from term grouping based on their overlapping level. (**B**) Overview pie-chart with functional groups. Statistical analysis was performed with the Cytoscape plugin ClueGo (v2.5.5) using the hypergeometric test and the following parameters: *p* < 0.01, kappa-score (k) = 0.5; (min/%) genes = 40/4%, GO tree levels: 13-15. Ontology databases: GOBP, GOCC, and GOMF, Reactome Pathways and KEGG.

**Table 1 cells-09-01666-t001:** List of validated NUP98-HOXA9 interaction partners found in the biotinylated fraction of NHA9-BioID.

Protein	Gene	NHA9-BioID	F.C.	Ref.
Nucleoporin NUP98	NUP98	+/+	−1.84	[46]
Chromosome region maintenance 1/Exportin 1	CRM1/XPO1	+/+	−2.13	[24,29]
mRNA export factor 1	RAE1	+/+	6.40	[46]
Nuclear factor of activated T cells	NFAT5	+/-	24.81	[24]
Histone-lysine N-methyltransferase 2A/Mixed lineage leukemia 1	KMT2A/ MLL1	+/+	−0.91	[28]
Host cell factor 1	HCFC1	+/+	−1.86	[27]
Histone deacetylase 1	HDAC1	+/+	−2.05	[47]
O-GlcNac transferase subunit p110	OGT	-/-		[27]
CREB binding protein	CREBBP/p300	-/-		[47,48]

F.C.—Fold change; +/+, present in both replicates; +/−, present in one of the replicates; −/−, not detected in any of in NHA9-BioID nor BirA^R118G^ replicates.

**Table 2 cells-09-01666-t002:** List of validated SET-NUP214 interaction partners found in the biotinylated fraction of SN214-BioID.

Protein	Gene	SN214-BioID	F.C.	Ref.
Nuclear RNA export factor 1	NXF1/ TAP	+/+	1.37	[25]
Chromosome region maintenance 1/Exportin 1	CRM1/XPO1	+/+	0.16	[23,25]
Nucleoporin 62	NUP62	+/+	1.79	[23]

F.C., Fold change; +/+, present in both replicates.

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
