# Peer review of "Disclosing the Interactome of Leukemogenic NUP98-HOXA9 and SET-NUP214 Fusion Proteins Using a Proteomic Approach"

_cells, 2020, doi:10.3390/cells9071666_

Round 1

Reviewer 1 Report

The manuscript demonstrates to study the two chimeras NUP98-HOXA9 and SET-NUP214 and their interaction as a determinant of cellular transformation and oncogenesis using acute leukemia cells. The authors have set forth the hypothesis that these fusion proteins have ability to inhibit nuclear export through interaction with epigenetic regulators. The authors conducted full interactome analysis and identified that both fusion proteins interact with major regulators of RNA processing and perturb the transcriptional program of the tumor suppressor p53, Wnt signaling, MAPK, and estrogen receptor and nuclear export receptor CRM1. Overall, the work is interesting, straightforward and the results support the conclusions. Although the concept of gene fusion in acute leukemia is not new and both fusion proteins have been studied in the past; however the proteomics approach undertaken is the highlight of the study.

Author Response

We thank the reviewer for the kind and positive evaluation of our manuscript.

Reviewer 2 Report

Comments in reading order:

Line 42 : NUP98-HOXA9, (chromosome translocation data)

Line 43: HOXA9 full name is missing

Line 46: NUP214 and SET (chromosome translocation data)

Line 55: DOT1L full name is missing.

Line 63: Materials and Methods contain very similar information as Appendix A 459 which should be clarified.

Line 64: It is claimed that all the experiments were carried out at room temperature unless otherwise specified. The total RNA isolation some aspects need four celcius which is missing.

Line 98: Pulldown of biotinylated proteins: One of the most critical techniques, which is incompletely described, an approximate cell number used for the lysis is not described. It is not known how much protein came down from the beads after digestion, was used for the western blot and mass spectrometry / samples.

How to explain that the enriched endogenous biotinylated proteins are not visible intensly in bound fraction on western blot.

Line 175: In the Figure 1 d, section of bound fraction, why are the endogenous enriched biotinylated proteins not visible?

Line 203: Table 1. It is shown that the CREBBP/p300 was not there, however the authors connect this protein with the WDR5 which was not listed in the table. What is the reason for that?

Line 249: To prove that NUP98-HOXA9 affects Wnt signalling pathway by carrying out immunoprecipitation for one or two candidates of Wnt signalling pathway would be great. That could strengthen the statement.

Line 289: Table S4 is missing.

Line 374 – 383: TFAP2A transcription factor has been highlighted by the authors, and as they explained it is shown in the FIGURE 2/3, however, that TFAP2A was not listed in those figures.

Line 426-429: It has been stated that for some known NUP98-HOXA9 and SET-NUP214 binding partners observed like the CRM1 and MLL1, there was no significant enrichment in NHA9-BioID and SN214-BioID fractions related to BIRa alone. In TABLE 1 (in the biotinylated fraction of NHA9-BioID), the authors presented a -2.13 value for CMR1 and -0.91 for MLL. In TABLE 2 (in the biotinylated fraction of SN214-BioID), the authors presented a 0.16 value for CMR1 but there was no data about MLL. With the term "no significant enrichment", it may suppose to be a negative value in each fraction, which is not true for the CMR1 and  the MLL was not listed in the second table.

Line 444: It should be pointed out and proven what would be the druggable target proteins which would increase the weight of the article.

Author Response

Responses to Reviewer 2

Line 42: NUP98-HOXA9, (chromosome translocation data)

       As suggested by the reviewer, the chromosome translocation is now denoted.

Line 43: HOXA9 full name is missing

       The full name is now included.

Line 46: NUP214 and SET (chromosome translocation data)

       The chromosome translocation is now included (line 47).

Line 55: DOT1L full name is missing

       The full name is now given (line 56).

Line 63: Materials and Methods contain very similar information as Appendix A 459 which should be clarified.

       We agree with the Reviewer that due to the same title of the sections in Material and Methods and the Appendix, clarity was lacking. We have are now included the entire protocol of immunofluorescence in Materials and Methods (line 122 onwards) and have specified the remaining section titles in the Appendix A (lines 591, 600, and 630).  

Line 64: It is claimed that all the experiments were carried out at room temperature unless otherwise specified. The total RNA isolation some aspects need four celcius which is missing.

We have now specified that the reagents for total RNA isolation were kept on ice for manipulation (line 561). All temperatures are explicitly stated at each step.

Line 98: Pulldown of biotnylated proteins: One of the most critical techniques, which is incompletely described, an approximate cell number used for the lysis is not described. It is not known how much protein came down from the beads after digestion, was used for the western blot and mass spectrometry / samples.

The amount of cells used for lysis are now described (line 135). Protein amounts after pull down were not quantified, instead the total volume of the eluate was used for analysis. This is now described in line 137 onwards.

How to explain that the enriched endogenous biotinylated proteins are not visible intensely in boundfraction on western blot.

Indeed, in the bound fraction, some protein bands are intensive than others. The main reason for low band intensity on the blots is the major strength of the bond between biotin and streptavidin. The strength of the bond makes a complete elution of biotinylated proteins from the streptavidin-coated beads difficult/largely impossible. Typically, a significant portion of biotinylated proteins is not eluted from streptavidin-coated beads. Thus, band intensity on the blot might not be correlated with protein abundance, but instead results from better elution of some proteins from the magnetic beads as compared to others. Nevertheless, the pull-down in Figure 1D clearly shows little to no enrichment of proteins in the bound fraction of BirA alone, whereas specific proteins can be observed both in the bound fractions of NUP98-HOXA9 and SET-NUP214, although the bands in the latter are less intense.

Line 175: In the Figure 1 d, section of bound fraction, why are the endogenous enriched biotinylated proteins not visible?

We are not sure what reviewer exactly means with endogenous enriched biotinylated proteins. But as stated in line 173 in the original manuscript and now line 245, the biotinylated NHA9 is likely presenting the strongest band migrating slightly above the 100 kDa protein marker band. The biotinylated SN214-BioID cannot be allocated due to the low intensity of the signal, but might correspond to the band migrating below the 250 kDa marker. For BirAR118G, no specific protein enrichment is expected, as the biotinylation mediated by the biotin ligase, alone is unspecific. We have nowspecified this in the results section, line 246 onwards.

Line 203: Table 1. It is shown that the CREBBP/p300 was not there, however the authors connect this protein with the WDR5 which was not listed in the table. What is the reason for that?

We did not try to make any connection between the p300 and the WDR5 proteins. Both are simply known interactors of NUP98-HOXA9, as reported in previous literature. p300 did not co-precipitate with any of the NUP98-HOXA9 replicates, hence it is denoted as -/- (p300 did, however, co-IP with SET-NUP214). WDR5 was not identified in any of the samples and we hypothesized (line 277) that it was not detected because the protein is mutated in these cells, and thus might not localize in proximity to the BioID fusion proteins.

Line 249: To prove that NUP98-HOXA9 affects Wnt signalling pathway by carrying out immunoprecipitation for one or two candidates of Wnt signalling pathway would be great. That could strengthen the statement.

We agree with the Reviewer that experimental validation of the candidates would strengthen our statements. However, this might not simply be done by immunoprecipitation, and we therefore prefer to keep validations for future studies.

Line 289: Table S4 is missing.

Table S3 is now included.

Line 374 – 383: TFAP2A transcription factor has been highlighted by the authors, and as they explained it is shown in the FIGURE 2/3, however, that TFAP2A was not listed in those figures.

TFAP2A is listed in Table S2, and not shown in Figure 2 and 3. We have clarified this in the text (line 457 on wards).

Line 426-429: It has been stated that for some known NUP98-HOXA9 and SET-NUP214 binding partners observed like the CRM1 and MLL1, there was no significant enrichment in NHA9-BioID and SN214-BioID fractions related to BIRa alone. In TABLE 1 (in the biotinylated fraction of NHA9-BioID), the authors presented a -2.13 value for CMR1 and -0.91 for MLL. In TABLE 2 (in the biotinylated fraction of SN214-BioID), the authors presented a 0.16 value for CMR1 but there was no data about MLL. With the term "no significant enrichment", it may suppose to be a negative value in each fraction, which is not true for the CMR1 and the MLL was not listed in the second table.

Tables 1 and 2 refer to previously validated interaction partners of either NUP98-HOXA9 or SET-NUP214 that were also found as interactors of NHA9-BioID and SN214-BioID. MLL is not included in Table 2, as it has not been validated as an interactor of SET-NUP214. We have specified this in the text. We agree that the term "no significant enrichment" may imply negative values in each fraction and have rephrased the text to "not strongly enriched" to avoid ambiguousness. All in lines 519-521.

Line 444: It should be pointed out and proven what would be the druggable target proteins which would increase the weight of the article.

       We have now named druggable targets (line 545), but we hope that the reviewer agrees that a proof will take several years of future research and is beyond the scope of the current manuscript.

Reviewer 3 Report

Mendes et al used proximity dependent biotin identification to study the interactors of NUP98-HOXA9 and SET-NUP214 fusion proteins. The results suggest that both fusion proteins interact with major regulators of RNA processing, with translation-associated proteins and also that both chimeras’ affect p53 transcriptional programme. Other cellular processes perturbed are specific to one of the two fusions. The study provides clues on interactors of nucleoporin fusion proteins and processes affected.

Major comments:

  1. The pattern of expression of SN214-BioID and of the GFP version really are that identical. The BioID version specks seem fewer and bigger.
  2. What is the difference between cells with foci of SET-NUP214 expression and those with a rim of its expression around the nuclear membrane? What is the proportion of the two populations? It there no biotinylation corresponding to the protein expressed at the nuclear rim?
  3. Can you speculate why there are 10 times more interactors of the SN214-BioID than of NHA9-BioID although the signal is more localised?
  4. What was the overlap of the two methods for NES identification? Why did you decide to do the GO ontology on one of them rather than on common NESs?
  5. I am afraid, I do not understand the meaning of negative F.C. in Table 1. Does it mean that there are more of these proteins in the BirA control pull-down? Does it count as a validation?
  6. A very short explanation of what proximity biotinylation approach for protein interactor identification may render the paper easier to understand for people who are not familiar with the system. Include a sentence on why you chose this approach r
  7. A figure of interactors common for the two fusion proteins and their locatisation/function etc would be useful for a quick take of what is similar.

Minor comments:

  1. Figures are normally referenced in the results section and not repeated in the discussion. It really disturbs the flow of reading.
  2. Gene nomenclature convention requires their names to be put in italics.
  3. LOUCY cells have a translocation or a deletion? It is not quite the same.
  4. You call NUP98-HOXA9-BirAR118G – NHA9-BioID or NUP98-HOXA9-BioID. It would be easier to follow, if just one name was used throughout. The same is true for the other fusion.

Author Response

Responses to Reviewer 3:

  1. The pattern of expression of SN214-BioID and of the GFP version really are that identical. The BioID version specks seem fewer and bigger.

We have not quantified the size and the frequency of the foci observed for SN214-BioID or GFP-SN214, but we have performed these transfections several times and have not detected substantial differences between the one and the other. Please see some other examples:

  1. What is the difference between cells with foci of SETNUP214 expression and those with a rim of its expression around the nuclear membrane? What is the proportion of the two populations? It there no biotinylation corresponding to the protein expressed at the nuclear rim?

We have not quantified the proportion of cells with  the one or the other pattern of localization. However, we observed that the presence of the fusion protein at the rim is likely less frequent than in the foci, or probably only less evident due to a weaker fluorescent intensity. We observed the rim localization also in patient-derived samples using gold immunolabelling and electron microscopy (unpublished data).

We expect that biotinylation occurs both at the foci and at the rim, when the fusion protein is localized there. This could, in part explain the finding of several NE-associated proteins in the SN214-biotinylated proteins (like laminA, some nucleoporins, etc).

  1. Can you speculate why there are 10 times more interactors of the SN214-BioID than of NHA9-BioID although the signal is more localised?

As we show in the beginning of the results section, SET-NUP214 is localised to nuclear foci and at the nuclear rim, whereas NUP98-HOXA9 is in the foci throughout the nucleus, likely bound to chromatin. This might implicate that SET-NUP214 is more prone to encounter NE and cytoplasmic proteins than NUP98-HOXA9. In fact, this might also explain why so many mitochondrial proteins are found among SN214-BioID biotinylated proteins. To have a definite answer to the question, it would be interesting to compare SN214 with NUP214 alone using BioID or other IP/MS approaches, such as GFP-TRAP, for example.

Another explanation might be the nature of the binding partner: whereas HOXA9 is a DNA-binding protein that targets NUP98-HOXA9 to the chromatin, SET is an epigenetic regulator that can inhibit the acetylation of histone and non-histone proteins. Thus, the group of SET interactors might include much more proteins than the group of HOXA9 interactors. It is, therefore, possible, that, because SET is almost completely preserved in the SET-NUP214, the fusion might interact with several of the endogenous SET interactors. This hypothesis is supported by the network of protein interactions generated from the BioGrid database (see Figures below).

Unique interactors of HOXA9 (BioGrid database (v3.5.187, assessed on 01.07.2020). Total of 20 unique interactors, corresponding to 29 interactions.

Unique interactors of SET (BioGrid database (v3.5.187, assessed on 01.07.2020). Total of 171 unique interactors, corresponding to 220 interactions

  1. What was the overlap of the two methods for NES identification? Why did you decide to do the GO ontology on one of them rather than on common NESs?

Both NES identification software (NES Finder 0.2 and LocNES) identify the classical NES necessary for CRM1 binding. NES peptides are usually 8-15 amino acids long with regularly spaced conserved hydrophobic residues (Ï•X2-3Ï•X2-3 Ï•X Ï•, Ï• corresponding to hydrophobic residues, typically leucine, methionine, valine, isoleucine or phenylalanine, X corresponding to any amino acid). The LocNES software integrates distinct parameters to identify classic NES. The software uses machine learning, namely the Support Vector Machine (SVM) model, that incorporates both sequence and biophysical features, such as predicted intrinsic disorder, secondary structure and solvent accessibilities [1,2].

The NES Finder software adopts a more conservative approach and identifies sequence motifs that follow the classic NES consensus [1,3]. In a first step, we used both software for prediction, to increase our confidence in the results. In a second step, we performed GO analysis using the results only from NES Finder 0.2, as a more conservative approach, as it has been reported that LocNES led to a higher rate of false positives  [4].

As suggested, we performed the GO analysis considering the common NES+ to the two softwares (see figure below) and obtained similar results as the ones described in the Results. However, when considering the common NES- between the two software, no significant enrichment was observed in any of the GO categories (data not shown).

Most represented cellular components among common NES+ identified by NES Finder 0.2 and Loc NES.

  1. I am afraid, I do not understand the meaning of negative F.C. in Table 1. Does it mean that there are more of these proteins in the BirA control pull-down? Does it count as a validation?

This is one of the major limitations of our BioID approach, in which we used transient transfection instead of stable, inducible expression of the BioID fusion proteins. Because the transfection efficiency is much better for the control BirA than for any of the two fusion proteins (NUP98-HOXA9 and SET-NUP214), the rate of biotinylation from BirA alone is expected to be higher than that of the fusions. This, together with the fact that this modified BirA (BirAR118G) is a promiscuous biotin ligase, likely results in higher amounts of biotinylated proteins in the control. For this reason, we employed a normalization strategy that would consider the differences in biotinylation that are simply due to the differences between the expression of the BioID fusion proteins and the control BirAR118G. After normalization, we performed GO on BirAR118G-biotinylated proteins, as described in the methods section (i.e., LFQNORMALIZED ≥ 0.6). This showed no enrichment in any of the GO categories tested (GOBP, GOMF and GOCC), supporting the idea that our control biotinylates endogenous proteins in an unspecific manner and, thus, is a suitable control for the experiment.

Moreover, given the differences in transfection efficiency between control and NHA9-BioID and SN214-BioID constructs, we concluded that our approach still fails in the identification of some potential candidates, because the control biotin ligase seems to induce higher biotinylation in known interactors, such as CRM1 and others. As a result, we reason that the main limitation of our strategy is the failure in the identification of known interactors.

The most “interesting” candidates, after normalization, are the ones showing the highest LFQNORMALIZED values. These proteins, albeit being biotinylated by BirA alone, are more enriched with NHA9 or SN214 (after the pulldown), meaning that their interaction with the fusion proteins is rather of specific nature.

  1. A very short explanation of what proximity biotinylation approach for protein interactor identification may render the paper easier to understand for people who are not familiar with the system. Include a sentence on why you chose this approach.

       As suggested by the Reviewer, a short explanation of the method and the choice in now included in the Introduction (line 61 onwards).

  1. A figure of interactors common for the two fusion proteins and their locatisation/function etc would be useful for a quick take of what is similar.

As suggested, a figure and a table (Figure S7 and Table S6) summarizing the common interactors are now included and referred to in the results section (line 442).

Minor comments:

  1. Figures are normally referenced in the results section and not repeated in the discussion. It really disturbs the flow of reading.

Referencing to figures in the Discussion is not uncommon, although we agree that it should be mainly occur in the Results section. We have removed the figure references in the discussion, except for references to supplementary tables and for some results that are explicitly mentioned in the Discussion and are relevant in the context of the Discussion.

  1. Gene nomenclature convention requires their names to be put in italics.

We put gene names in italics, where we have previously missed this: in the plasmid section in Material and methods (line 78 onwards) and in Figure S". Throughout the manuscript, we used the nomenclature convention for human genes and proteins: GENE, TRANSCRIPT, PROTEIN.

  1. LOUCY cells have a translocation or a deletion? It is not quite the same.

       Loucy cells carry del9(q34.11; q34.13), which is now been clarified in the text (line 80).

  1. You call NUP98-HOXA9-BirAR – NHA9-BioID or NUP98-HOXA9-BioID. It would be easier to follow, if just one name was used throughout. The same is true for the other fusion.

       As suggested by the Reviewer, we are now using exclusively NHA9-BioID and SN214-BioID throughout the manuscript. The designations NUP98-HOXA9 and SET-NUP214 are used when referring to the fusion proteins in general.

References mentioned in the rebuttal:

  1. Xu, D.; Marquis, K.; Pei, J.; Fu, S.-C.; CaÄŸatay, T.; Grishin, N.V.; Chook, Y.M. LocNES: a computational tool for locating classical NESs in CRM1 cargo proteins. Bioinformatics 2014, 31, 1357-1365, doi:10.1093/bioinformatics/btu826.
  2. Fu, S.-C.; Imai, K.; Horton, P. Prediction of leucine-rich nuclear export signal containing proteins with NESsential. Nucleic acids research 2011, 39, e111-e111, doi:10.1093/nar/gkr493.
  3. Fung, H.Y.; Fu, S.C.; Brautigam, C.A.; Chook, Y.M. Structural determinants of nuclear export signal orientation in binding to exportin CRM1. eLife 2015, 4, doi:10.7554/eLife.10034.
  4. Lee, Y.; Pei, J.; Baumhardt, J.M.; Chook, Y.M.; Grishin, N.V. Structural prerequisites for CRM1-dependent nuclear export signaling peptides: accessibility, adapting conformation, and the stability at the binding site. Scientific reports 2019, 9, 6627, doi:10.1038/s41598-019-43004-0.

Round 2

Reviewer 2 Report

MS is now ready to be published.

Author Response

Thank you!